# Functional Characterization of 5-*O*-Glycosyltranferase Transforming 3-*O* Anthocyanins into 3,5-*O* Anthocyanins in *Freesia hybrida*

**DOI:** 10.3390/ijms26104542

**Published:** 2025-05-09

**Authors:** Tingting Bao, Xiang Zheng, Yicong Pang, Ruifang Gao, Xiaotong Shan, Shirui Zhu, Shadrack Kanyonji Kimani, Xiang Gao, Yueqing Li

**Affiliations:** 1Key Laboratory of Molecular Epigenetics of MOE, Northeast Normal University, Changchun 130024, China; botanist.adnan@yahoo.com (A.); baott372@nenu.edu.cn (T.B.); zhengxiang@nenu.edu.cn (X.Z.); pangyicong@nenu.edu.cn (Y.P.); gaorf203@nenu.edu.cn (R.G.); lizishanxt@163.com (X.S.); zhushirui214@nenu.edu.cn (S.Z.); skanyonji@gmail.com (S.K.K.); gaoxiang424@163.com (X.G.); 2Shenzhen Branch, Guangdong Laboratory of Lingnan Modern Agriculture, Key Laboratory of Synthetic Biology, Ministry of Agriculture and Rural Affairs, Agricultural Genomics Institute at Shenzhen, Chinese Academy of Agricultural Sciences, Shenzhen 518120, China

**Keywords:** anthocyanidin, flavonoid, flower color, modification genes, MBW complex

## Abstract

Floral coloration in *Freesia hybrida* is predominantly attributed to anthocyanins, with glycosylation playing a critical role in their stability and diversity. This study investigates the molecular mechanisms underlying color variation between *F. hybrida* cultivars, focusing on anthocyanin 5-*O*-glucosyltransferases (An5GTs). HPLC analysis revealed that ‘Pink Passion’ petals accumulate 3,5-*O*-diglucosidic anthocyanins, absent in ‘Red River^®^’ and ‘Ambiance’. RNA-seq identified seven candidate *Fh5GT* genes, with phylogenetic and subcellular localization analyses confirming their classification as cytosolic glycosyltransferases. Expression profiling highlighted elevated transcript levels of *Fh5GT1*, *Fh5GT3*, and *Fh5GT7* in ‘Pink Passion’, correlating with its di-glucosidic anthocyanin accumulation. In vitro enzymatic assays demonstrated that Fh5GT3 and Fh5GT7 preferentially glucosylate 3-*O*-monoglucosidic anthocyanins to form stable 3,5-*O*-diglucosides, with minimal activity on anthocyanidins to generate 5-*O*-glucosidic anthocyanins. Heterologous expression of *Fh5GT3* and *Fh5GT7* in *Arabidopsis* complemented anthocyanin deficiency in *5gt* mutants, restoring pigmentation. These findings elucidate the potential role of 5GTs in modulating floral color diversity through anthocyanin modification, providing insights for targeted breeding strategies to enhance ornamental traits in horticultural species.

## 1. Introduction

Floral coloration results from the accumulation of pigments, mostly composed of flavonoids, carotenoids, and betalains. Flavonoids which are water-soluble phenolic compounds, encompass chalcones, flavones, flavonols, flavanols, anthocyanins, and proanthocyanidins [1,2]. Among these, anthocyanins stand out as the most significant pigments that are widely distributed across the plant kingdom. These pigments play a key role in determining the coloration of different plant tissues and organs, resulting in hues such as red, blue, and purple in petals and fruits [3,4]. Furthermore, they play an essential role in various biological processes, such as pollination, seed dissemination, and protection against ultraviolet radiation [5,6]. Moreover, anthocyanins are valuable as food additives and provide several health benefits, including antioxidant activity, free radical scavenging, antibacterial and antiviral properties, cardiovascular disease prevention, and protective effects against liver damage and cancer [7].

Anthocyanins are characterized by a C6-C3-C6 flavonoid backbone, consisting of a benzopyran heterocyclic ring (C ring), a fused aromatic ring (A ring), and a phenyl substituent (B ring). Anthocyanin biosynthesis originates within the flavonoid branch of the phenylpropanoid metabolic pathway and has been widely investigated across various plant species, such as *Zea mays*, *Petunia hybrida*, *Arabidopsis thaliana*, and *Antirrhinum majus* [8,9]. Typically, the anthocyanin biosynthetic pathway begins with the condensation of 4-coumaroyl-CoA and three molecules of malonyl-CoA, catalyzed by chalcone synthase (CHS), leading to the formation of naringenin chalcone. This compound is then isomerized to naringenin by chalcone isomerase (CHI), establishing the basic C6-C3-C6 backbone. Subsequent hydroxylation at the C3 position on the C ring by flavanone 3-hydroxylase (F3H) generates dihydrokaempferol, which can be further hydroxylated by either flavonoid 3′ hydroxylase (F3′H) or flavonoid 3′5′-hydroxylase (F3′5′H) on ring B to produce dihydroquercetin and dihydromyricetin, respectively. Dihydrokaempferol, dihydroquercetin and dihydromyricetin are then converted into anthocyanidins through sequential catalysis by dihydroflavonol reductase (DFR) and anthocyanidin synthase (ANS). The primary anthocyanidins include pelargonidin, cyanidin, delphinidin, peonidin, and malvidin. It is noteworthy that anthocyanidins are inherently unstable and are typically glycosylated to form more stable anthocyanins. The initial gylcosylation of anthocyanidins is catalyzed by anthocyanidin 3-*O*-glucosyltransferase (3GT or An3GT), which transfers a glucoside molecule from UDP-glucoside to the C3 position on the C ring of anthocyanidin [10]. Subsequent modifications of anthocyanins, encompassing glycosylation, methylation, and acylation, along with variations in the number, type and position of these modification groups, contribute significantly to the extensively natural diversity of anthocyanins. While the fundamental anthocyanin biosynthetic pathway has been well characterized [11,12,13], the details of the sequential modifications of anthocyanins remain relatively unexplored. Additionally, the biosynthesis and accumulation of anthocyanins are tightly regulated at multiple levels, with transcriptional regulatory networks playing a central role. In many plant species, the MBW (MYB-bHLH-WD40) complex, composed of MYB transcription factors, bHLH proteins, and WD40 repeat proteins, acts as a key regulator of anthocyanin biosynthesis [6,14,15]. However, the regulatory mechanisms governing various anthocyanin-modifying enzymes and their contributions to floral color diversity remain largely unexplored.

Glycosylation is the most prominent modification of anthocyanins and plays an important role in contributing to the complexity and diversity of these pigments. Typically, mono-glucosidic anthocyanins, which are produced in plants through the action of anthocyanidin 3-*O*-glucosyltransferase, can be converted into di-glucosidic anthocyanins with the help of anthocyanin 5-*O*-glucosyltransferase (5GT or An5GT) [16,17]. An5GTs have been functionally characterized in some plant species, including *Gentiana triflora* and *Cyclamen purpurascens* [18,19]. However, whether 5GTs can directly modify anthocyanidins and how they affect final flower color remains largely unresolved.

*Freesia hybrida*, a monocotyledonous ornamental species, is popular worldwide as a cut flower due to its vibrant colors, distinctive fragrance, and long vase life. Its flowers exhibit a range of colors, including purple, red, pink, yellow, and white, with varied pigmentation patterns. Previous investigations in *F. hybrida* ‘Red River^®^’, which produces red flowers, identified five types of C3 position glucosylated anthocyanins, while no anthocyanins were detected in *F. hybrida* ‘Ambiance’ with white petals [20,21]. Moreover, key enzymes involved in primary anthocyanin biosynthesis, including four glucosyltransferases, have also been functionally characterized [20,22,23]. In this study, the anthocyanins in another cultivar, *F. hybrida* ‘Pink Passion’ which produces pink petals were analyzed. Comparatively, the pink petals of *F. hybrida* ‘Pink Passion’ exhibited a higher accumulation of di-glucosidic anthocyanins. The key 5GT enzymes responsible for di-glucosidic anthocyanin biosynthesis, as well as their catalytic capacities on both anthocyanidins and 3-*O*-glucosylated anthocyanins, were investigated. Furthermore, in vivo overexpression assays involving *5GTs* in *Arabidopsis* confirmed their roles in the biosynthesis of di-glucosidic anthocyanins. Our results highlight the significant contribution of 5GTs to flower color and lay a foundation for further molecular design aimed at flower color improvement in ornamental plants.

## 2. Results

### 2.1. Different Colored F. hybrida Cultivars Accumulate Varied Anthocyanins

Our previous study has identified delphinidin 3-*O*-glucoside, cyanidin 3-*O*-glucoside, petunidin 3-*O*-glucoside, peonidin 3-*O*-glucoside, and malvidin 3-*O*-glucoside as the dominant anthocyanins present in the petals of *F. hybrida* ‘Red River^®^’ [20]. To compare anthocyanin profiles across different *Freesia* cultivars, *F. hybrida* ‘Red River^®^’, ‘Pink Passion’ and ‘Ambiance’ with distinct petal colors were employed (Appendix A). Anthocyanin analysis indicated that there were five types of mono-glucosidic anthcyanins in *F. hybrida* ‘Red River^®^’ petals, while no detectable anthocyanins were accumulated in *F. hybrida* ‘Ambiance’ petals (Figure 1A and Appendix A). In contrast, *F. hybrida* ‘Pink Passion’ petals possessed all the primary anthocyanins found in *F. hybrida* ‘Red River^®^’ except cyanidin 3-*O*-glucoside. Additionally, four modified anthocyanins, probably delphinidin 3,5-*O*-diglucoside, cyanidin 3,5-*O*-diglucoside, petunidin 3,5-*O*-diglucoside, and malvidin 3,5-*O*-diglucoside were detected in ‘Pink Passion’ petals (Figure 1B). To investigate the accumulation patterns of anthocyanins, the development of *Freesia* flowers was categorized into five stages (Appendix A) [24]. Our analysis revealed that the levels of most anthocyanins increased with flower development (Figure 1C,D), suggesting a corresponding increment in the expression of anthocyanin biosynthetic genes.

Interestingly, there are more 3,5-*O*-diglucosidic anthocaynins in the petals of *F. hybrida* ‘Pink Passion’ compared to ‘Red River^®^’. The differences in anthocyanin profiles between these two cultivars may result from variations in the sequence or expression levels of candidate *5GTs*.

### 2.2. The Difference in Expression Levels, Rather Than Sequence Variations of 5GTs, May Account for the Differing Accumulation of 3,5-O-Diglucosidic Anthocyanins Between F. hybrida ‘Red River^®^’ and ‘Pink Passion’

To identify the candidate 5GTs responsible for the biosynthesis of 3,5-*O*-diglucosidic anthocyanins, the fully bloomed flowers of *F. hybrida* ‘Red River^®^’, ‘Pink Passion’ and ‘Ambiance’ were subjected for RNA-seq analysis. Based on the RNA-seq data, genes associated with anthocyanin biosynthesis were analyzed across these cultivars. Relative FPKM values were compared to assess variations in expression. Notably, white-flowered *F. hybrida* ‘Ambiance’ exhibited negligible expression of anthocyanin biosynthesis genes, consistent with its lack of pigmentation. Both ‘Red River^®^’ and ‘Pink Passion’ showed high expression of these genes, confirming active anthocyanin biosynthesis (Appendix A). Moreover, expression levels of MBW components correlated with anthocyanin accumulations across these cultivars, supporting their regulatory roles (Appendix A).

Using the An5GT sequences from *Iris hollandica* [25] and *Gentiana triflora* [18] as query sequences in a BLAST (v. 2.13.0) search against the *Freesia* transcriptome database, we identified seven putative anthocyanin-related *5GT* genes, which included the previously published *Fh5GT1* and *Fh5GT2* [22]. The remaining genes were designated as *Fh5GT3* to *Fh5GT7*. Phylogenetic analysis revealed that all seven Fh5GTs clustered within the anthocyanin 5-*O*-glucosyltransferases subclade, confirming their classification as candidate 5GT proteins (Figure 2A). The seven candidate *Fh5GT* sequences were cloned from both *F. hybrida* ‘Red River^®^’ and ‘Pink Passion’. Sequence comparison analysis indicated that Fh5GTs from *F. hybrida* ‘Pink Passion’ shared identical sequences with those from *F. hybrida* ‘Red River^®^’ except for Fh5GT6, which exhibited only one residue difference between the two cultivars (Appendix A). Moreover, amino acid sequence alignments demonstrated that all Fh5GTs contained the plant secondary product glycosyltransferase (PSPG) motif, suggesting their potential roles in the modification of anthocyanins (Appendix A).

Typically, anthocyanins are synthesized in the cytosol and subsequently transported into the vacuole [26]. To explore the possible subcellular localization of these candidate Fh5GTs, the coding sequences of all Fh5GTs were seamlessly fused with GFP and transiently transformed into *Arabidopsis* leaf protoplasts. Results showed that Fh5GTs-GFP fusion proteins localized to the cytosol, whereas free GFP was diffusely distributed throughout the protoplast (Figure 2B). The results indicate that the modification of anthocyanins probably occurs in the cytosol.

As reported in our previous studies, the anthocyanin content in *Freesia* increases during flower development, primarily accumulating in the petals [21,22,23,27,28]. Notably, the di-glucosidic anthocyanins were only detected in *F. hybrida* ‘Pink Passion’ (Figure 1 and Appendix A). To assess the correlation between anthocyanin accumulation and candidate *Fh5GTs*, the expression patterns of all *Fh5GTs* were examined in samples collected at various flowering stages, tissues, and cultivars. Interestingly, while the expression levels of all *Fh5GTs* increased during flowering and peaked at full bloom, *Fh5GT1*, *Fh5GT3*, and *Fh5GT7* exhibited higher expression levels compared to the others (Figure 3A). Generally, *Fh5GTs* were most highly expressed in petals and stamens, with the exception of *Fh5GT5*, which showed elevated expression levels in pistils and stamens (Figure 3B). Remarkably, only *Fh5GT1*, *Fh5GT3*, *Fh5GT5*, and *Fh5GT7* displayed relatively higher expression levels in *F. hybrida* ‘Pink Passion’ compared to *F. hybrida* ‘Red River^®^’ and ‘Ambiance’. Overall, these findings suggest that the elevated expression levels of *Fh5GT1*, *Fh5GT3*, and *Fh5GT7* in *F. hybrida* ‘Pink Passion’ may be pivotal in determining the abundance of di-glucosidic anthocyanins.

### 2.3. Fh5GTs Prefer 3-O-Glucosidic Anthocyanins Instead of Anthocyanidins as Acceptors

Since Fh5GT1 and Fh5GT2 have previously been characterized for their roles in modifying anthocyanin biosynthesis, the functions of the remaining Fh5GTs were further investigated through in vitro enzymatic assays. Recombinant Fh5GT proteins were first extracted and purified from bacteria (Appendix A). Subsequently, these recombinant proteins were incubated with UDP-glucoside and various 3-*O*-glucosidic anthocyanins. The results indicated that both Fh5GT3 and Fh5GT7 effectively modified the primary 3-*O*-glucosidic anthocyanins into their corresponding 3,5-*O*-diglucoside forms: delphinidin 3,5-*O*-diglucoside, cyanidin 3,5-*O*-diglucoside, petunidin 3,5-*O*-diglucoside, peonidin 3,5-*O*-diglucoside, and malvidin 3,5-*O*-diglucoside (Figure 4). Notably, the residual difference observed between Fh5GT6 from ‘Red River^®^’ and Fh5GT6 from ‘Pink Passion’ did not affect its enzymatic activity toward these anthocyanins (Figure 4). To determine whether these Fh5GTs could directly add glucosides to anthocyanidins, recombinant Fh5GT proteins were incubated with UDP-glucoside and cyanidin. However, only minimal peaks were observed in the reactions involving Fh5GT3 and Fh5GT7, which may be cyanidin 5-*O*-glucoside (Figure 4 and Appendix A). Overall, these findings suggest that Fh5GTs can modify 5 position of 3-*O*-glucosidic anthocyanins or anthocyanidins, with a clear preference for 3-*O*-glucosidic anthocyanins as substrates.

### 2.4. Fh5GT3 and Fh5GT7 Complement the Anthocyanin Deficiency in Arabidopsis 5gt Mutant

To further investigate the roles of Fh5GT3 and Fh5GT7, these two *Fh5GT* genes were overexpressed in an *Arabidopsis 5gt* mutant, which exhibits low anthocyanin levels. Kanamycin-selected T2 transgenic seedlings were germinated and grown on half-strength MS medium with 3% sucrose. Phenotypic analysis of the transgenic lines revealed a distinct purple coloration in the hypocotyls and cotyledons (Figure 5A). RT-PCR (reverse transcription PCR) was used to confirm the expression of *Fh5GTs* genes in the T2 transgenic lines (Figure 5B). The transgenic *Arabidopsis* exhibited detectable *Fh5GT* amplicons, and the anthocyanin levels were significantly restored in the transformed plants (Figure 5C). In conclusion, the results suggest that both Fh5GT3 and Fh5GT7 are bona fide anthocyanin 5-*O*-glycosyltransferases, demonstrating their crucial roles in anthocyanin biosynthesis in plants.

## 3. Discussion

Anthocyanins are a type of flavonoids responsible for the red, purple, and blue colors found in many fruits, vegetables, and flowers. The biosynthesis and regulation of primary anthocyanins have been extensively investigated, revealing a high degree of conservation among various plant species [6,29]. Beyond the biosynthesis of primary anthocyanin backbones, modifications such as glycosylation, methylation, and acylation play pivotal roles in enhancing both the stability and diversity of anthocyanins [30]. For example, research on modification genes such as GTs (particularly 5GTs and 7GTs) are largely lagged, probably due to two main factors: First, the complex nature of anthocyanin modification involves at least 6 glycosyl donors -glucosyl, galactosyl, rhamnosyl, xylosyl, arabinosyl and galacturonyl-affecting at least 6 positions (3, 5, 7, 3′, 4′ and 5′) on various aglycons. Furthermore, modifications can interact with previous glycone modifications [31]. Second, unlike biosynthetic genes involved in anthocyanin production, the loss-of-function or overexpression of a single modification gene typically results in minimal observable phenotypic changes. These modifications of anthocyanins may fine-tune color rather than introduce significant alterations, which could be pivotal for specific plants in attracting particular pollinators or facilitating plant-environment interactions [32].

The abundant presence of flavonoids composed of flavonols, anthocyanins, and proanthocyanidins in flowers makes *Freesia* an excellent model for investigating flavonoid biosynthesis and regulation within monocots. In the current study, we primarily focus on two questions: (1) What differences in anthocyanin profiles exist among different Freesia varieties? (2) How do GTs, particularly 5GTs, function among *Freesia* varieties with different colors? The varying anthocyanin accumulation patterns observed in *F. hybrida* ‘Red River^®^’, ‘Pink Passion’, and ‘Ambiance’ provide ideal systems to explore these questions. In particular, *F. hybrida* ‘Pink Passion’ accumulates both 3- and 5-position modified anthocyanins, in addition to those detected in *F. hybrida* ‘Red River^®^’ (Figure 1). Anthocyanin 5-*O*-glucosyltransferases, such as IhAn5GT and AtAn5GT, which glycosylate anthocyanidin 3-*O*-glucosides to produce anthocyanin 3,5-*O*-diglucosides [17,25] provid promising baits for identifying *Freesia* An5GTs. Phylogenetic analysis grouped Fh5GT1–Fh5GT7 with other An5GTs, indicating their potential roles in modifying anthocyanidin 3-*O*-glucosides (Figure 2A). Sequence analysis revealed the presence of the PSPG motif shared by 3,5-*O*-glycosyltransferases [33] in Fh5GT1–Fh5GT7. Previous studies have indicated that plant glycosyltransferases are typically localized in cytoplasm [34] and our findings confirm that Fh5GTs localize in the cytoplasm (Figure 2B). In *Vitis amurensis* and *Gentiana. triflora*, the expression levels of *Va5GT* and *Gt5GT7* correlate with anthocyanin accumulation during berry skin ripening and flower development [18,35]. Likewise, *F. hybrida* follows this trend, with *Fh5GT1*, *Fh5GT3*, and *Fh5GT7* being expressed throughout all stages of flower development, peaking in fully pigmented petals (Figure 3). In addition to the previously characterized Fh5GT1 and Fh5GT2, in vitro (Figure 4) and in vivo (Figure 5) functional characterizations of Fh5GTs further confirmed the roles of Fh5GT3 and Fh5GT7 in modifying 3-*O*-anthocyanidins. Overall, Fh5GT1, Fh5GT2, Fh5GT3, and Fh5GT7 are bona fide 5-*O*-glucosyltraferases in *F. hybrida* flowers.

In contrast to 3GTs, which transfers UDP-activated sugar moieties to the 3-OH position of anthocyanidins, An5GT specifically glycosylates anthocyanins at the 5-*O*-position, leading to the production of highly stable anthocyanidin 3,5-diglucosides [36,37]. It is generally understood that anthocyanidins first undergo glycosylation at the 3-position, followed by subsequent 5-position glycosylation. Despite the identification of numerous *An5GT* genes involved in this process [38], to our knowledge, no reports have confirmed that *An5GT* directly modifies anthocyanidins into either anthocyanidin 3-*O*-glucodise or anthocyanidin 3,5-*O*-diglucosides. To verify the specificity of An5GTs, cyanidin was used as a substrate along with UDP-glucose in the presence of Fh5GTs, where Fh5GT3 and Fh5GT7 were able to convert cyanidin to cyanidin 5-*O*-glucoside, albeit with low efficiency (Figure 4). This finding confirms that anthocyanin 5-*O*-glycosyltransferases exclusively transfer sugar to the 5-position of anthocyanins, providing the possibility for future production of specialized anthocyanins.

In addition to the modification of anthocyanins, anthocyanin biosynthesis is tightly regulated through various transcription factors [39]. The primary regulatory complex involved in this process is the MYB-bHLH-WD40 (MBW) complex, which consists of three key components: (1) MYB transcription factors that bind to specific DNA sequences to regulate gene expression involved in anthocyanin production. (2) bHLH (basic Helix-Loop-Helix) proteins that cooperate with MYB proteins to enhance gene activation. (3) WD40 repeat proteins that serve as co-factors, stabilizing interactions between MYB and bHLH proteins, thereby facilitating the formation of a functional MBW complex. The accumulation of MBW complexes substantially activates the transcription of anthocyanin biosynthesis genes, leading to increased anthocyanin accumulation [21,27]. However, the low transcripts of anthocyanin pathway genes, along with the resulting white petals of *F. hybrida* (Appendix A), may result from the deficiency of functional MBW complex [21,27]. In contrast, petals from *F. hybrida* ‘Pink Passion’ contained all the primary anthocyanins found in *F. hybrida* ‘Red River^®^’, except for cyanidin 3-*O*-glucoside (Figure 1). Four modified anthocyanins, delphinidin 3,5-*O*-diglucoside, cyanidin 3,5-*O*-diglucoside, petunidin 3,5-*O*-diglucoside, and malvidin 3,5-*O*-diglucoside, were identified in ‘Pink Passion’ petals (Figure 1). Moreover, *Fh5GT1*, *Fh5GT3*, and *Fh5GT7* exhibited significantly higher expression levels in *F. hybrida* ‘Pink Passion’ compared to ‘Red River^®^’ and ‘Ambiance’ (Figure 3). These results suggest that the elevated expression of *Fh5GT1*, *Fh5GT3*, and *Fh5GT7* in ‘Pink Passion’ is critical for the accumulation of di-glucosidic anthocyanins, likely contributing to the color variation observed between ‘Red River^®^’ and ‘Pink Passion’ (Figure 6). Although robust evidence is lacking, we hypothesize two possibilities: First, the expression of *Fh5GT1*, *Fh5GT3*, and *Fh5GT7* may operate independently of MBW complex regulation in *F. hybrida* ‘Red River^®^’. Second, the promoters of *Fh5GT1*, *Fh5GT3*, and *Fh5GT7* in *F. hybrida* ‘Red River^®^’ may mutate when compared to those in ‘Pink Passion’, leading to less efficient expressions of these glucosyltranferases.

This study investigates the genetic and metabolic processes underlying the color differences between *F. hybrida* ‘Red River^®^’ and *F. hybrida* ‘Pink Passion’ flowers. These differences are attributed to the presence of additional 3,5-*O*-diglucosidic anthocyanins in ‘Pink Passion’, which may finally contribute to its unique color. The diglycosylation of anthocyanins is mediated by glycosyltransferase enzymes (Fh5GT1, Fh5GT3 and Fh5GT7), which are highly expressed in ‘Pink Passion’. These findings enhance our understanding of anthocyanin production and modification by An5GTs, emphasizing their role in floral pigmentation and presenting potential applications in plant breeding and biotechnology for developing flowers with specific hues.

## 4. Materials and Methods

### 4.1. Plant Material and Growth Conditions

*F. hybrida* ‘Red River^®^’, ‘Pink Passion’, and ‘Ambiance’ were grown in a greenhouse at Northeast Normal University, with a controlled temperature of 15 °C and a photoperiod of 14 h light and 10 h dark. The development of *Freesia* flowers was divided into five stages (Stage 1: Buds < 10 mm in length, unpigmented. Stage 2: Buds 10–20 mm in length, slight pigmentation visible. Stage 3: Buds 20–30 mm in length, fully pigmented but unopened. Stage 4: Fully pigmented flowers prior to complete opening. Stage 5: Fully opened flowers with mature pigmentation.), and fully bloomed flowers were separated into calyx, petal, pistil, stamen, and torus, following our previously published papers [24]. *Arabidopsis thaliana* was planted in a greenhouse at a constant temperature of 22 °C, with a light exposure of 16 h followed by 8 h of darkness. All samples were harvested, immediately immersed in liquid nitrogen, and stored at −80 °C for long-term storage.

### 4.2. Anthocyanin Extraction and HPLC Analysis

For anthocyanin extraction, 0.15 g of plant samples were treated with 1 mL of 0.1% acidic methanol solution (HCl/MeOH, 1/999, *v*/*v*) at 4 °C for 12 h in the dark. The resultant extract was centrifuged at 12,000 rpm for 10 min, and the supernatant was filtered through a 0.22 μm membrane prior to HPLC analysis. 20 μL aliquot of the solution was analyzed using a Shimadzu HPLC system equipped with an ACCHROM XUnion 5-μm C18 column (250 × 4.6 mm) at 35 °C. The column was eluted with solvent systems A (0.1% HCl in H_2_O) and B (acetonitrile). The condition for HPLC analysis was as follows: 0–20 min, 10–11%B; 20–25 min, 50%B; 25–30 min, 50%B; 30–35 min, 10%B with a flow rate of 1 mL min^−1^. Anthocyanins were detected at 520 nm. Subsequently, all observed anthocyanins were identified based on the retention times of anthocyanin standard samples (Sigma Aldrich, St. Louis, MO, USA). To determine the content of a specific anthocyanin relative to the total anthocyanins, the formula: “Percentage of Anthocyanin = (Peak Area of Specific Anthocyanin/Total Peak Areas of All Anthocyanins) × 100%” was employed. To compare the levels of specific anthocyanins at different developmental stages, a One-Way Analysis of Variance (ANOVA) followed by Ducan’s test was employed.

### 4.3. RNA-Seq Analysis

For RNA-seq analysis, fully bloomed flowers of *F. hybrida* ‘Red River^®^’, ‘Pink Passion’ and ‘Ambiance’ were collected and sent to Biomarker Technologies (Beijing, China). Transcriptome analysis was performed on the illumina platform according to the company’s standard procedures. De novo transcriptome assembly was performed using the classical Trinity approach, which includes Inchworm, Chrysalis, and Butterfly [40]. The raw RNA-seq data have been deposited in the National Genomics Data Center (NGDC) under the BioProject accession number PRJCA038991.

### 4.4. Gene Cloning and Sequence Analysis

To isolate the candidate unigenes involved in anthocyanin biosynthesis, the plugin “Quick Find Best Homology” integrated in TBtools-II [41] was employed to screen the yielded *F. hybrida* transcriptome database. To identify anthocyanin-related *5GTs* in *F. hybrida,* a homology-based BLAST analysis was conducted on the *F. hybrida* transcriptome database in TBtools-II [41], using an E-value thresholod of 1e-5. An5GT sequences from *Iris hollandica* [25] and *Gentiana triflora* [17] were used as bait sequences. The identified sequences were initially annotated using the National Center for Biotechnology Information (NCBI) BLAST to predict coding regions. Specific primers (Appendix A) were designed with the purpose of cloning the candidate *5GT* genes, which were subsequently inserted into the p*ESI-Blunt* vector (Hieff Clone Zero TOPO-Blunt Cloning Kit, Yeasen, Shanghai, China) for sequence confirmation (Appendix A).

Multiple sequence alignments were performed using the cloned sequences alongside other glucosyltransferases from various plants. The sequences were processed using the online Clustal Omega algorithm (accessed on 1 December 2022, https://www.ebi.ac.uk/Tools/msa/clustalo/) with default parameters. For phylogenetic analysis, a neighbor-joining tree was constructed in MEGA X, with robustness assessed through bootstrap resampling analysis (1000 replicates).

### 4.5. Subcellular Localization Analysis

To investigate the subcellular localizations of Fh5GTs, the coding sequences of *Fh5GTs* were seamlessly cloned into the *Nde*I and *Cla*I digested *35S:LfPAP1-GFP* vector [42] using the Minerva Super Fusion Cloning Kit (US EverbrightR Inc., Suzhou, China), generating *35S:Fh5GTs-GFP*. The vectors were transformed into bacteria, and plasmids were extracted using the GoldHi EndoFree Plasmid Midi Kit (CWBIO, Beijing, PRC). Extracted plasmids were further concentrated by isopropanol and NaCl before protoplast transfection. Arabidopsis protoplasts were isolated from rosette leaves and transfected with different plasmids using PEG3350 following established methods [24,43]. After an incubation of 22 h, the protoplasts were observed using a laser scanning confocal microscopy.

### 4.6. Heterologous Expression of Fh5GTs in E. coli BL21

Prokaryotic fusion proteins were prepared according to previous protocols [43,44]. Briefly, the *Fh5GTs* genes were seamlessly cloned into pET-32a using the Minerva Super Fusion Cloning Kit (US EverbrightR Inc., Suzhou, China) and subsequently transformed into *E. Coli BL21* (*DE3*) cells. Recombinant proteins were induced by isopropyl-β-d-thiogalactopyranoside (IPTG) at 16 °C for 18 h. Following induction, the bacterial cells were harvested by centrifugation and lysed through sonication. The supernatant containing the recombinant proteins was pelleted and resuspended in phosphate-buffered saline (pH 7.4). The protein solution was further purified by Ni Sepharose columns (Sangon Biotech Co., Ltd., Shanghai, China). Routine SDS-PAGE analysis was performed to verify the purity of the purified proteins.

### 4.7. Enzymatic Assay

The enzymatic assays of Fh5GTs were performed following earlier described protocols with some modifications [35,45]. In brief, the standard reaction mixture comprised 100 µM anthocyanidins or 3-*O*-glucosidic anthocyanins as the glucosyl acceptor, 10 mM UDP-glucose as the glucosyl donor, and 30 µg of recombinant protein making up a total volume of 200 µL in 100 mM potassium phosphate buffer at pH 8.0. The reaction solution was incubated at 30 °C for 5 min and terminated by adding 50 µL of 5% HCl solution. Afterward, the samples were centrifuged at 12,000 rpm for 5 min, and supernatants were filtered through a 0.22 µm membrane filter before HPLC analysis. The glycosylation products were determined by comparing peaks to standard compounds.

### 4.8. Heterologous Expression of Fh5GTs in Arabidopsis

To investigate the functions of *Fh5GTs* in vivo, the CDS sequences of Fh5GT3 and Fh5GT7 were seamlessly subcloned into *Nde*I and *Cla*I digested in the backbone of *pUC19* vector to construct *35S:HA-Fh5GT3* and *35S:HA-Fh5GT7*. The constructs were transformed into *E. coli DH5α*, plated on Amp-resistant solid LB medium, and screened for positive clones. Plasmid DNA was extracted, digested with *EcoR*I, and ligated into binary *pPZP211* vector. The resulting constructs were introduced into *Agrobacterium tumefaciens* GV3101 via freeze–thaw method and used to transform Arabidopsis *5gt* mutant (At4g14090, Salk_108458, Columbia background) using the floral dip method [46]. Kanamycin-selected T2 transgenic seedlings were germinated and grown on 1/2 MS medium with 1% sucrose for phenotype observation. The RNA was extracted and reverse-transcribed into cNDA to assess the presence of *Fh5GTs*. Total anthocyanin was extracted from samples with 0.1% acidic methanol solution. After an overnight incubation, the absorbance values at wavelengths of 530 nm (A530) and 657 nm (A657) were recorded, and the relative anthocyanin content was calculated as A530 − 0.25 × A657.

## 5. Conclusions

This study elucidates the pivotal role of 5-O-glucosyltransferases (Fh5GTs) in modulating floral coloration in *Freesia hybrida* by enhancing anthocyanin stability and diversity. Comparative analysis of anthocyanin profiles across cultivars revealed that ‘Pink Passion’ accumulates 3,5-*O*-diglucosidic anthocyanins, absent in ‘Red River^®^’ and ‘Ambiance’, correlating with its distinct pink hue. RNA-seq identified seven Fh5GT candidates, with phylogenetic and localization analyses confirming their classification as cytosolic glycosyltransferases. Expression profiling highlighted elevated transcript levels of *Fh5GT1*, *Fh5GT3*, and *Fh5GT7* in ‘Pink Passion’, aligning with di-glucosidic anthocyanin accumulation. In vitro assays demonstrated their preference for 3-O-monoglucosidic anthocyanins, catalyzing the formation of stable 3,5-O-diglucosides, with minimal activity on anthocyanidins. Heterologous expression of *Fh5GT3* and *Fh5GT7* in Arabidopsis *5gt* mutants restored anthocyanin levels, confirming their functional roles. These findings underscore that differential expression of *Fh5GTs*, rather than sequence variation, drives color variation between cultivars. This study provides critical insights into the enzymatic basis of anthocyanin modification, offering a foundation for targeted breeding strategies to manipulate floral pigmentation in ornamental plants. By linking specific glycosyltransferases to color phenotypes, this work advances our ability to engineer vibrant and stable flower colors, enhancing horticultural value and ecological adaptability.

## Figures and Tables

**Figure 1 ijms-26-04542-f001:**
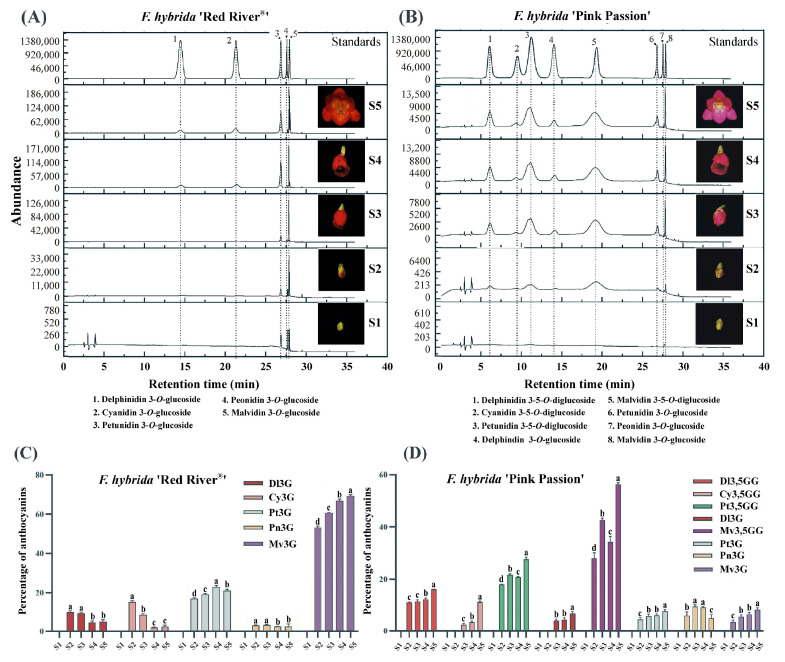
Anthocaynin analysis in petals of *F. hybrida* ‘Red River^®^’ and ‘Pink Passion’. HPLC analysis of anthocyanins in the petals of *F. hybrida* ‘Red River^®^’ (**A**) and ‘Pink Passion’ (**B**). S1-S5 represent five developmental stages. Bars represent 1.0 cm. The relative anthocyanin contents such as Dl3G (delphinidin 3-*O*-glucoside), Cy3G (cyanidin 3-*O*-glucoside), Pt3G (petunidin 3-*O*-glucoside), Pn3G (peonidin 3-*O*-glucoside) and Mv3G (malvidin 3-*O*-glucoside) in different flower development stages of *F. hybrida* ‘Red River^®^’ (**C**). The relative contents of Dl3,5GG (delphinidin 3,5-*O*-diglucoside) Cy3,5GG (cyanidin 3,5-*O*-diglucoside) Pt3,5GG (petunidin 3,5-*O*-diglucoside), Dl3G (delphinidin 3,5-*O*-diglucoside), Mv3,5GG, (malvidin 3,5-*O*-diglucoside), Pt3G (petunidin 3-*O*-glucoside) and Mv3G (malvidin 3-*O*-glucoside) in different flower development stages of *F. hybrida* ‘Pink Passion’ (**D**). One-way ANOVA was carried out to compare statistical differences (Ducan, *p* < 0.05).

**Figure 2 ijms-26-04542-f002:**
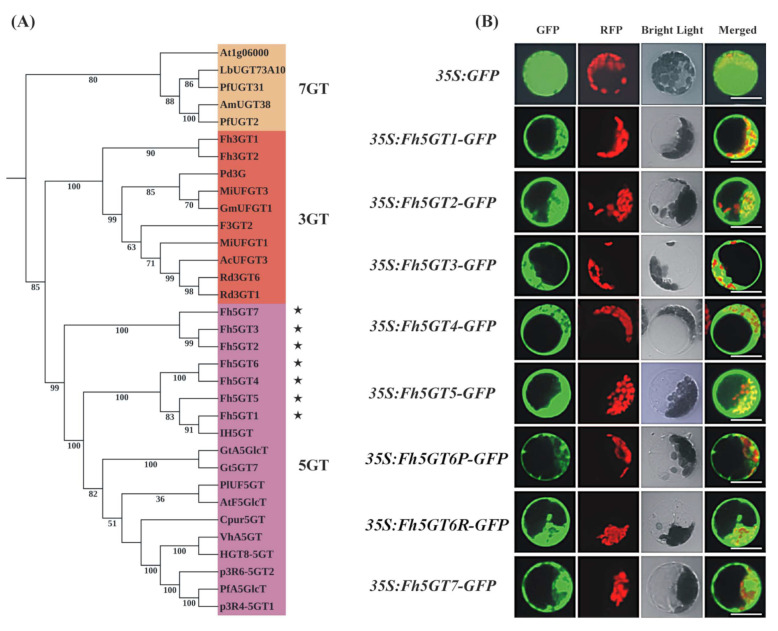
Phylogenetic and subcellular localization analysis of Fh5GTs. (**A**) Neighbor joining phylogenetic analysis of Fh5GTs with the glycosyltransferases from other plants. Stars represent candidate 5GT sequences from *F. hybrida*. Genbank accession numbers: *Paeonia lactiflora* PlUF5GT (JQ070807), *Cyclamen purpurascens* Cpur5GT (LC597018), *Arabidopsis thaliana* AT4G14090 (NP_193146.1), At1g06000 (BT006579.1), *Gentiana triflora* Gt5GT7 (B2NID7.1), GtA5GlcT (BAG32255.1), *Iris hollandica* IH5GT (Q767C8.1), *Perilla frutescens* p3R4-5GT1 (Q9ZR27.1), p3R6-5GT2 (Q9ZR26.1), HGT8-5GT (Q9ZR25.1), PfA5GlcT (BAA36421.1), PfUGT31 (BAG31952), PfUGT2 (BAG31951.1), *Verbena hybrida* VhA5GT (BAA36423.1), *Freesia hybrida* Fh3GT1 (ADK75021.1), Fh3GT2 (MK945761), *Actinidia chinensis* AcUFGT3a (A0A2R6Q8R5.1), F3GT2 (A0A2R6P624.1), *Garcinia mangostana* GmUFGT1 (ACM62748), *Mangifera indica* MiUFGT1 (BBJ35509.1), MiUFGT3 (BBJ35510.1), *Rhododendron delavayi* Rd3GT1 (P0DO58.1), Rd3GT6 (P0DO59), *Paeonia delavayi* Pd3G (AQZ26785.1), *Antirrhinum majus* AmUGT38 (BAG16513.1), *Lycium barbarum* LbUGT73A10 (BAG80536.1). (**B**) Subcellular localization of Fh5GTs. The plasmids Fh5GT1-GFP, Fh5GT2-GFP, Fh5GT3-GFP, Fh5GT4-GFP, Fh5GT5-GFP, Fh5GT6P-GFP, Fh5GT6R-GFP and Fh5GT7-GFP were introduced into protoplasts obtained from the rosette leaves of 3–4 week old Arabidopsis plants. The transfected protoplasts were then incubated in darkness at room temperature for 20–22 h. GFP fluorescence was observed and recorded using a confocal microscope. Scale bar = 25 μm. Gene names ending with “P” indicate genes from *F. hybrida* ‘Pink Passion’ and those ending with “R” indicate genes from *F. hybrida* ‘Red River^®^’.

**Figure 3 ijms-26-04542-f003:**
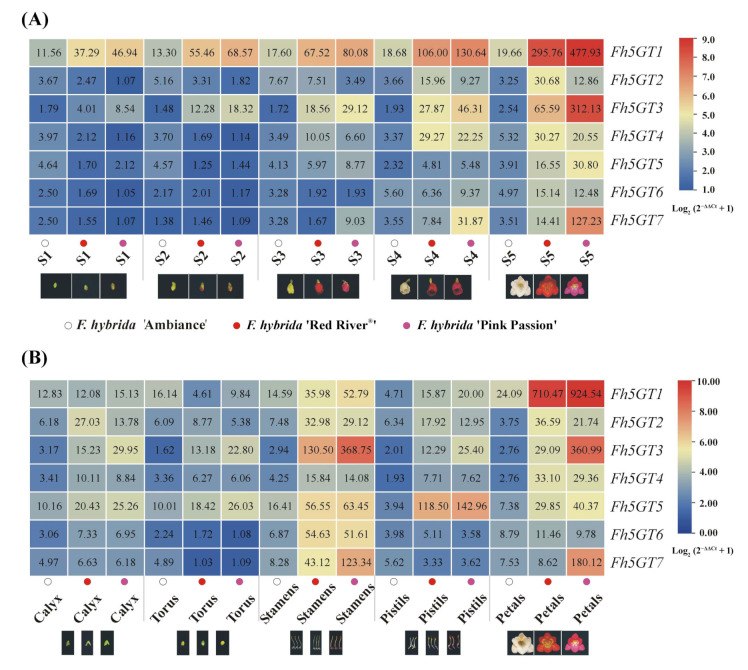
Spatiotemporal expression analysis of Fh5GTs. (**A**) Expression analysis of *Fh5GTs* at different flower developmental stages of *F. hybrida* ‘Ambiance’, *F. hybrida* ‘Red River^®^’ and *F. hybrida* ‘Pink Passion’. (**B**) Expression analysis of *Fh5GTs* in different tissues of *F. hybrida* ‘Ambiance’, *F. hybrida* ‘Red River^®^’ and *F. hybrida* ‘Pink Passion’. The mean data from RT-qPCR (reverse transcription quantitative PCR) of at least three biological replicates were normalized as log_2_ (2^−ΔΔCt^ + 1).

**Figure 4 ijms-26-04542-f004:**
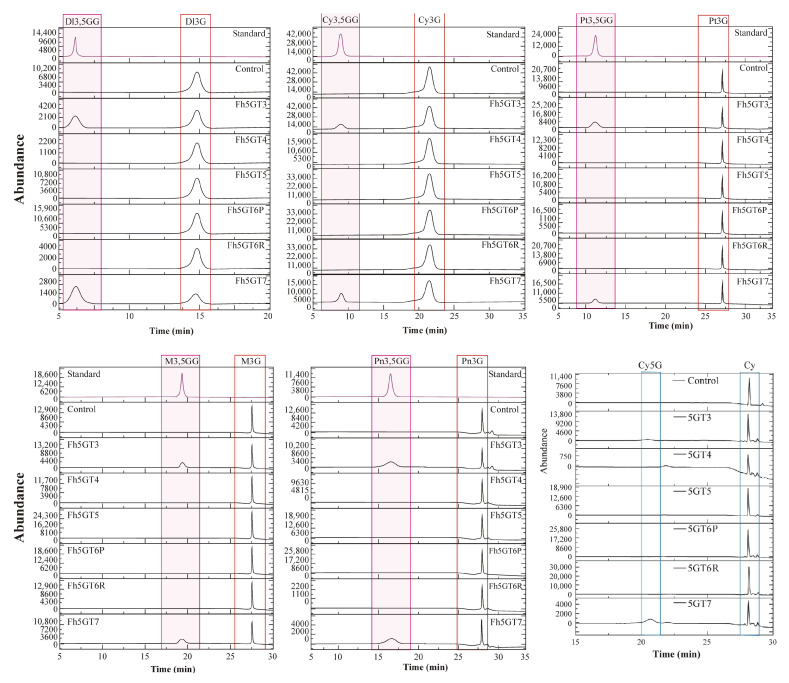
In vitro enzymatic assays of Fh5GTs toward various anthocyanins. HPLC analysis of the activities of recombinant Fh5GTs toward delphinidin 3-*O*-diglucoside, cyanidin 3-*O*-diglucoside, petunidin 3-*O*-diglucoside, peonidin 3-*O*-diglucoside, and malvidin 3-*O*-diglucoside in the presence of UDP-glucose. Dl3G, delphinidin 3-*O*-diglucoside; Cy3G, cyanidin 3-*O*-diglucoside; Pt3G, petunidin 3-*O*-diglucoside; Pn3G, peonidin 3-*O*-diglucoside; Mv3G, malvidin 3-*O*-diglucoside; Dl3,5GG, delphinidin 3,5-*O*-diglucoside, Cy3,5GG, cyanidin 3,5-*O*-diglucoside; Pt3,5GG, petunidin 3,5-*O*-diglucoside; Pn3,5GG, peonidin 3,5-*O*-diglucoside; Mv3,5GG, malvidin 3,5-*O*-diglucoside. Control represented the standard substrates reacting with proteins extracted from bacteria expressing empty vector.

**Figure 5 ijms-26-04542-f005:**
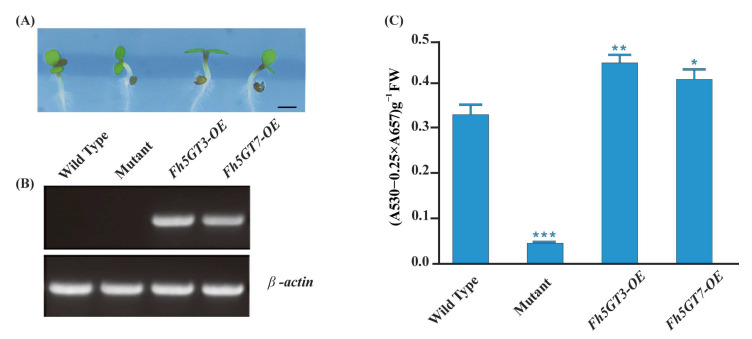
Pigment complementation of *Arabidopsis 5gt* mutant seedlings with *Fh5GT3* and *Fh5GT7*. (**A**) Phenotypes of wild-type, mutant, and transgenic *Arabidopsis* seedlings. Bars represent 1.0 mm. (**B**) The expression of *Fh5GT* genes detected by reverse transcription polymerase chain reaction in wild-type, mutant and transgenic lines. (**C**) The anthocyanin contents in wild-type, mutant, and transgenic *Arabidopsis* seedlings. Data represented mean ± SD of three biological replicate. Statistically significant differences between mutant and wild-type as well as transgenic plants are indicated by asterisks, as determined by Student’s *t*-test (* *p* < 0.05; ** *p* < 0.01; *** *p* < 0.001).

**Figure 6 ijms-26-04542-f006:**
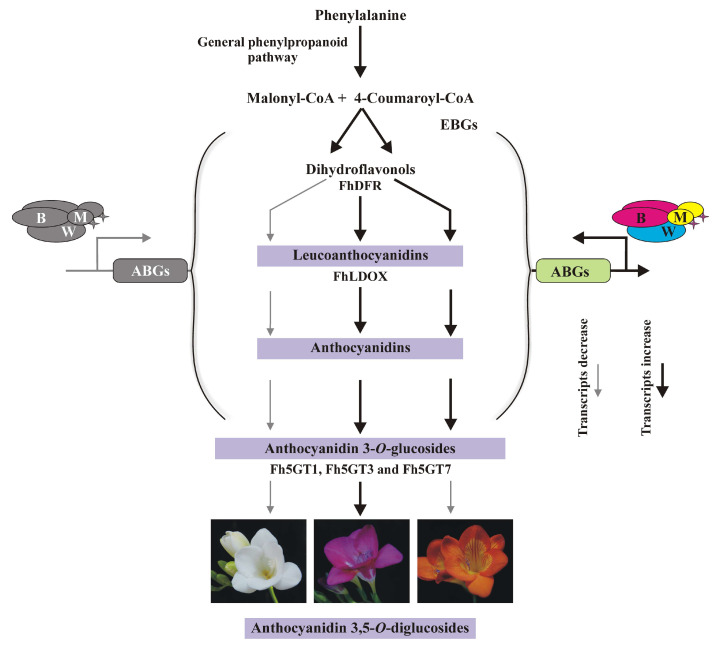
Proposed molecular mechanism underlying anthocyanin biosynthesis of ‘Red River^®^’, ‘Pink Passion’ and ‘Ambiance’. In ‘Ambiance’, the low expression of the MBW complex leads to low expression levels of anthocyanin biosynthesis genes (ABGs), culminating in white flowers due to reduced anthocyanin biosynthesis. In ‘Red River^®^’ and ‘Pink Passion’, the MBW complex likely enhances the transcription of ABGs, resulting in petal pigmentation. Moreover, the highly expressed Fh5GT1, Fh5GT3, and Fh5GT7 catalyze the conversion of anthocyanidin 3-*O*-glucosides into anthocyanidin 3,5-*O*-diglucosides, which may contribute to color variation. EBGs, early biosynthetic genes which typically includes CHS, CHI, F3H, F3′H. ABGs, anthocyanin biosynthetic genes.

## Data Availability

Data are contained within the article or Appendix A. For other information, please contact the corresponding author.

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
