# Peer review of "Functional Characterization of 5-O-Glycosyltranferase Transforming 3-O Anthocyanins into 3,5-O Anthocyanins in Freesia hybrida"

_ijms, 2025, doi:10.3390/ijms26104542_

Round 1

Reviewer 1 Report

Comments and Suggestions for Authors The study mainly analyzes anthocyanin content differences in petals across different F. hybrida cultivars and investigates the role of 5GT gene in catalyzing 3-O anthocyanins to form 3,5-O anthocyanins. This is an interesting study with valuable findings. However, several issues remain in the manuscript: -Line 134: Authors mentioned conducting RNA-seq analysis on three cultivars, yet no relevant descriptions or citations appear in the Results section. Please clarify and add corresponding analyses. It is suggested to include differentially expressed genes (DEGs) in the anthocyanin biosynthesis pathway. Are major differences mainly observed in this pathway? Subsequent revisions should be made in Abstract, Discussion, and Conclusion sections accordingly. Additionally, raw data should be deposited to public databases, and methodological descriptions require more details. -Authors emphasized expression level differences rather than sequence variations in 5GTs across three species. While the Discussion extensively addresses transcription factors, no related results were presented despite available RNA-seq data. It is recommended to at least supplement expression data to enrich the manuscript content given their RNA-seq dataset. -Line 100: Authors referred to their previous analysis of F. hybrida 'Red River®' petals. Upon checking the cited reference, we found it also analyzed five developmental stages. Please clarify whether data used in this manuscript for 'Red River®' matches previous findings. If so, this should be explicitly stated in figures, methods, etc. -Line 104: Total ion chromatogram (TIC) for F. hybrida 'Ambiance' petals appears missing in Figure 1A. -Figure 4 needs clearer annotations specifying which compounds each protein reacts with in each panel. Current title notes are insufficient for readers. Additionally, controls should be clearly labeled, as control groups typically contain substrate (3-O-glucosidic anthocyanins) without enzyme, while products should be validated with authentic standards. Further clarification is needed. -Figure 5A shows unclear anthocyanin accumulation in transgenic cotyledons. Scale bars should be added. In Figure 5C, wild-type controls should be used for comparative analysis, so the ** symbol on wild-type bars should be reassigned to mutant bars. -Figure 6 requires revision. While Fh5GTs functions were characterized, expression patterns of entire pathway genes remain unclear. Whether these specific Fh5GTs directly determine color differences among cultivars needs further verification. -Resolution of Figure 1A, B, and Supplementary Figure S1 needs improvement. -All petal images are too small and lack scale bars. Authors should add a comparative figure of petals from three cultivars across five stages at higher resolution in main or supplementary figures. -All "-O-" in anthocyanin structures and statistical "P" values should be italicized.

Author Response

The study mainly analyzes anthocyanin content differences in petals across different F. hybrida cultivars and investigates the role of 5GT gene in catalyzing 3-O anthocyanins to form 3,5-O anthocyanins. This is an interesting study with valuable findings. However, several issues remain in the manuscript.

Response: We sincerely thank the reviewer for their constructive feedback and for recognizing the significance of our study.

-Line 134: Authors mentioned conducting RNA-seq analysis on three cultivars, yet no relevant descriptions or citations appear in the Results section. Please clarify and add corresponding analyses. It is suggested to include differentially expressed genes (DEGs) in the anthocyanin biosynthesis pathway. Are major differences mainly observed in this pathway? Subsequent revisions should be made in Abstract, Discussion, and Conclusion sections accordingly. Additionally, raw data should be deposited to public databases, and methodological descriptions require more details.

Response:We sincerely appreciate the reviewer’s insightful feedback and the opportunity to clarify our RNA-seq analysis. While we generated a de novo transcriptome assembly, the lack of a genome-complete protein database may limit the accuracy of GO or KEGG interpretations of DEGs. To avoid speculative conclusions, we intentionally restricted our analysis to identifying candidate 5GT genes, which aligned with the study’s primary goal. Broader transcriptomic comparisons (e.g., GO or KEGG analysis of DEGs across cultivars) will be reported in a separate manuscript focused on genome-wide regulatory networks in Freesia. However, in response to the reviewer’s suggestion, we analyzed the expression profiles of key anthocyanin biosynthesis genes (e.g., CHS, DFR, ANS) across the three cultivars. The results are now included in Supplementary Figure S3. Notably, F. hybrida 'Ambiance' (white petals) showed significantly lower expression of these genes compared to pigmented cultivars ('Red River®' and 'Pink Passion'), consistent with its lack of anthocyanin accumulation. Additionally the raw RNA-seq data for the three Freesia hybrida cultivars ('Red River®', 'Pink Passion', and 'Ambiance') have been deposited in the National Genomics Data Center (NGDC) under the BioProject accession number PRJCA038991. This information has been added to the Materials and Methods section.

-Authors emphasized expression level differences rather than sequence variations in 5GTs across three species. While the Discussion extensively addresses transcription factors, no related results were presented despite available RNA-seq data. It is recommended to at least supplement expression data to enrich the manuscript content given their RNA-seq dataset.

Response: We sincerely thank the reviewer for their valuable feedback. Based on our RNA-seq data, we analyzed the expression of key anthocyanin biosynthesis genes and MBW components across the three cultivars. The results, now included in Supplementary Figure S3 and described in Result Section. Accordingly, we expanded the Introduction to include a brief discussion of MBW complex regulating anthocyanin biosynthesis. We also removed extraneous speculation about MBW-mediated regulation and refocused the Discussion on the critical roles of Fh5GTs in diversifying anthocyanin structures.

-Line 100: Authors referred to their previous analysis of F. hybrida 'Red River®' petals. Upon checking the cited reference, we found it also analyzed five developmental stages. Please clarify whether data used in this manuscript for 'Red River®' matches previous findings. If so, this should be explicitly stated in figures, methods, etc.

Response: We thank the reviewer for their careful attention to detail and for raising this important point. While the developmental stage classifications (e.g., five stages of flower maturation) align with our earlier study on F. hybrida 'Red River®', the specific datasets (e.g., anthocyanin quantification, gene expression profiles) are derived from independent biological replicates and analytical workflows. The consistency between our current findings and prior results (e.g., anthocyanin types in 'Red River®') underscores the reliability of the experimental system. However, this study extends beyond previous work by: Analyzing two additional cultivars ('Pink Passion' and 'Ambiance'), Identifying novel 5GT genes and characterizing their enzymatic functions, Linking glycosylation activity to phenotypic variation.

-Line 104: Total ion chromatogram (TIC) for F. hybrida 'Ambiance' petals appears missing in Figure 1A.

Response: Thank you so much for your valuable suggestion. We truly appreciate your attention to detail. In response, we have included the HPLC result of F. hybrida 'Ambiance' petals in the Supplementary Figure S2. As no anthocaynins were detected in 'Ambiance' fully-opened flowers, we did not analyze its developmental stages. We hope this clarifies the situation, and we are grateful for your feedback.

-Figure 4 needs clearer annotations specifying which compounds each protein reacts with in each panel. Current title notes are insufficient for readers. Additionally, controls should be clearly labeled, as control groups typically contain substrate (3-O-glucosidic anthocyanins) without enzyme, while products should be validated with authentic standards. Further clarification is needed.

Response: Thank you for your careful work. In response to your suggestion, we have now clearly labeled the control groups and the products, ensuring they are validated with authentic standards. We truly appreciate your valuable suggestion, which has greatly helped us improve the manuscript.

-Figure 5A shows unclear anthocyanin accumulation in transgenic cotyledons. Scale bars should be added. In Figure 5C, wild-type controls should be used for comparative analysis, so the ** symbol on wild-type bars should be reassigned to mutant bars.

Response:Thank you for your thoughtful suggestion. We have added the scale bars to Figure 5A for clarity, and wild-type controls have been incorporated for the comparative analysis in Figure 5C, with the ** symbol now reassigned to the mutant bars. Your input was very helpful!

-Figure 6 requires revision. While Fh5GTs functions were characterized, expression patterns of entire pathway genes remain unclear. Whether these specific Fh5GTs directly determine color differences among cultivars needs further verification.

Response: We have revised Figure 6 and adjusted the Figure Legend and Discussion to emphasize that Fh5GTs likely contribute to color differences by diversifying anthocyanin glycosylation, while acknowledging the need for further validation of their direct role.

-Resolution of Figure 1A, B, and Supplementary Figure S1 needs improvement. -All petal images are too small and lack scale bars. Authors should add a comparative figure of petals from three cultivars across five stages at higher resolution in main or supplementary figures.

Response: Thank you for your work. We have replaced all images in Figure 1A, B, and Supplementary Figure S1 (now Figure S2) with high-resolution versions. Scale bars are now added to a comparative panel showing petal phenotypes across three cultivars at five developmental stages as a new Supplementary Figure S1.

-All "-O-" in anthocyanin structures and statistical "P" values should be italicized.

Response: Thank you for your professional comment. We have italicized all "-O-" in the anthocyanin structures and the statistical "P" values as requested. Once again, thank you for your valuable feedback, which will undoubtedly enhance the quality of our manuscript. We are grateful for your helpful guidance and thoughtful consideration.

Reviewer 2 Report

Comments and Suggestions for Authors

The manuscript entitled “Functional characterization of 5-O-glycosyltransferase transforming 3-O anthocyanins into 3,5-O anthocyanins in Freesia hybrida” presents a well-structured and comprehensive study on the floral pigmentation mechanism based on anthocyanin pigments.

This study is well-designed, integrating transcriptomic, biochemical, and functional approaches to elucidate the role of anthocyanin 5-O-glucosyltransferases (5GTs) in the coloration process of Freesia hybrida. The combination of RNA-seq, phylogenetic analysis, subcellular localization, and enzymatic assays provides strong evidence for the functional roles of the identified Fh5GTs.

The research is significant in clarifying the process of anthocyanin modification throughout flower development via the activity of 5GTs. The findings offer clear applications in the molecular breeding of ornamental plants.

After careful consideration of the manuscript, I recommend its acceptance following minor revisions concerning formatting, presentation, and typographical errors.

  • Statistical analyses should be provided for the results shown in Figures 1C and 1D to clearly demonstrate the significant differences between the compared groups.
  • The source of the anthocyanin standards should be clearly stated.
  • The classification of floral developmental stages from S1 to S5 should be described in more detail rather than simply being cited in the text.
  • In the extraction and HPLC method section (Section 4.2), the quantification method for anthocyanins should be specified, including the concentration range of standards, calibration curve equation, LOD, and LOQ.
  • Supplementary Figure S1 does not clearly show representative peaks of the detected anthocyanins. Therefore, it is not necessary to include this figure.

Author Response

The manuscript entitled “Functional characterization of 5-O-glycosyltransferase transforming 3-O anthocyanins into 3,5-O anthocyanins in Freesia hybrida” presents a well-structured and comprehensive study on the floral pigmentation mechanism based on anthocyanin pigments.

This study is well-designed, integrating transcriptomic, biochemical, and functional approaches to elucidate the role of anthocyanin 5-O-glucosyltransferases (5GTs) in the coloration process of Freesia hybrida. The combination of RNA-seq, phylogenetic analysis, subcellular localization, and enzymatic assays provides strong evidence for the functional roles of the identified Fh5GTs.

The research is significant in clarifying the process of anthocyanin modification throughout flower development via the activity of 5GTs. The findings offer clear applications in the molecular breeding of ornamental plants.

After careful consideration of the manuscript, I recommend its acceptance following minor revisions concerning formatting, presentation, and typographical errors.

Response: Thank you very much for your thoughtful and constructive feedback. We greatly appreciate your positive comments regarding the study. In response, we have made revisions to address the formatting, presentation, and typographical errors. We believe these changes enhance the clarity and quality of the manuscript. 

Statistical analyses should be provided for the results shown in Figures 1C and 1D to clearly demonstrate the significant differences between the compared groups.

Response: Thank you for your comment. Following you suggestion, we have provided the analysis.

The source of the anthocyanin standards should be clearly stated.

Response:Thank you for your reminder. We have provided the information.

The classification of floral developmental stages from S1 to S5 should be described in more detail rather than simply being cited in the text.

Response: Thank you for your suggestion. We have provided a new supplementary Figure S1 to show the developmental stages. Additionally, we described more detail in the methods.

In the extraction and HPLC method section (Section 4.2), the quantification method for anthocyanins should be specified, including the concentration range of standards, calibration curve equation, LOD, and LOQ.

Response: Thank you for your comment. Following your suggestion, we have provided the details in the latest manuscript.

Supplementary Figure S1 does not clearly show representative peaks of the detected anthocyanins. Therefore, it is not necessary to include this figure.

Response: We appreciate your attention to the clarity of Supplementary Figure S1 (Figure S2 in latest manuscript). However, we respectfully request to retain this figure, as it serves a critical purpose in demonstrating the absence of detectable anthocyanins in F. hybrida 'Ambiance' petals. Again, we are grateful for your helpful guidance and thoughtful consideration.

Round 2

Reviewer 1 Report

Comments and Suggestions for Authors I am pleased to observe that the authors have largely addressed the majority of the issues raised. However, there remain a few points that require the authors' attention. -The supplementary figures incorporated by the authors do not appear to be cited within the main text, nor have I found any descriptions pertaining to these supplementary figures. Please ensure that both appropriate citations and detailed descriptions are added. -Figure S3: It seems that certain gene families relevant to the anthocyanin biosynthesis pathway are absent from the figure, such as F3’5’H, F3’H, BZ1, UGT75C1, and so forth. Were these genes not differentially expressed in the transcriptome analysis? Alternatively, was their identification overlooked? -Meanwhile, the authors are requested to include a dedicated 'Conclusions' section in the manuscript, summarizing the key findings, implications, and broader contributions of the study to the field.

Author Response

I am pleased to observe that the authors have largely addressed the majority of the issues raised. However, there remain a few points that require the authors' attention. -The supplementary figures incorporated by the authors do not appear to be cited within the main text, nor have I found any descriptions pertaining to these supplementary figures. Please ensure that both appropriate citations and detailed descriptions are added. -Figure S3: It seems that certain gene families relevant to the anthocyanin biosynthesis pathway are absent from the figure, such as F3’5’H, F3’H, BZ1, UGT75C1, and so forth. Were these genes not differentially expressed in the transcriptome analysis? Alternatively, was their identification overlooked? -Meanwhile, the authors are requested to include a dedicated 'Conclusions' section in the manuscript, summarizing the key findings, implications, and broader contributions of the study to the field.

Response: We sincerely appreciate the reviewer’s acknowledgment of our revisions and are grateful for the opportunity to address the remaining points.

  • Supplementary figures citations.

All supplementary figures and tables are explicitly cited and described in the main text. We have double-checked the manuscript to ensure proper referencing.

  • Gene coverage in Figure S3 (Anthocyanin Pathway).

F3’H and F3’5’H: These genes were not initially included as the pathway focused on core biosynthetic genes (e.g., CHS, DFR, ANS). However, in response to the reviewer’s request, we have now added F3’H and F3’5’H expression profiles to Supplementary Figure S3.

BZ1 (3GT) and UGT75C1 (5GT): BZ1 homolog was included in the original pathway. UGT75C1 homologs, while identified in our transcriptome, were excluded from the figure as they are central to this study and investigated in detail in the following parts of the manuscript.

  • Addition of a conclusion section.

Although IJMS does not mandate a Conclusion section, we have added one to the revised manuscript to concisely summarize key findings, implications for horticultural breeding, and future research directions.

Again, we appreciate the reviewer’s careful work.